# AI Applications to Reduce Loneliness Among Older Adults: A Systematic Review of Effectiveness and Technologies

**DOI:** 10.3390/healthcare13050446

**Published:** 2025-02-20

**Authors:** Yuyi Yang, Chenyu Wang, Xiaoling Xiang, Ruopeng An

**Affiliations:** 1Division of Computational and Data Sciences, McKelvey School of Engineering, Washington University in St. Louis, St. Louis, MO 63130, USA; 2Department of Surgery, Division of Public Health, Washington University in St. Louis, St. Louis, MO 63130, USA; chenyu.wang@wustl.edu; 3School of Social Work, University of Michigan, Ann Arbor, MI 48109, USA; xiangxi@umich.edu; 4Silver School of Social Work, New York University, New York, NY 10003, USA; ra4605@nyu.edu

**Keywords:** artificial intelligence, loneliness, older adults, social robots, mental health

## Abstract

**Background/Objectives:** Loneliness among older adults is a prevalent issue, significantly impacting their quality of life and increasing the risk of physical and mental health complications. The application of artificial intelligence (AI) technologies in behavioral interventions offers a promising avenue to overcome challenges in designing and implementing interventions to reduce loneliness by enabling personalized and scalable solutions. This study systematically reviews the AI-enabled interventions in addressing loneliness among older adults, focusing on the effectiveness and underlying technologies used. **Methods:** A systematic search was conducted across eight electronic databases, including PubMed and Web of Science, for studies published up to 31 January 2024. Inclusion criteria were experimental studies involving AI applications to mitigate loneliness among adults aged 55 and older. Data on participant demographics, intervention characteristics, AI methodologies, and effectiveness outcomes were extracted and synthesized. **Results:** Nine studies were included, comprising six randomized controlled trials and three pre–post designs. The most frequently implemented AI technologies included speech recognition (*n* = 6) and emotion recognition and simulation (*n* = 5). Intervention types varied, with six studies employing social robots, two utilizing personal voice assistants, and one using a digital human facilitator. Six studies reported significant reductions in loneliness, particularly those utilizing social robots, which demonstrated emotional engagement and personalized interactions. Three studies reported non-significant effects, often due to shorter intervention durations or limited interaction frequencies. **Conclusions:** AI-driven interventions show promise in reducing loneliness among older adults. Future research should focus on long-term, culturally competent solutions that integrate quantitative and qualitative findings to optimize intervention design and scalability.

## 1. Introduction

Loneliness is a subjective feeling of being alone, arising from the gap between an individual’s desired and actual social connections [1]. It affects people of all ages, including older adults. According to the World Health Organization (WHO), 20–34% of older adults report loneliness in countries such as the United States and across Europe [2,3]. Strong evidence links loneliness to increased risks of physical and mental health problems in older adults, resulting in reduced quality of life and life expectancy [4,5,6,7]. The health impact of chronic loneliness is comparable to smoking 15 cigarettes a day [6]. Recognizing its profound public health implications, researchers and policymakers have intensified efforts to develop and test interventions to reduce loneliness and mitigate its effects [8].

Current strategies for reducing loneliness typically fall into four categories: (1) improving social skills, (2) enhancing social support, (3) increasing opportunities for social contact, and (4) addressing maladaptive social cognition [9]. However, these interventions have shown mixed effectiveness, often facing challenges in scalability due to limited resources and inconsistent outcomes across diverse populations [9,10]. These limitations highlight the need for innovative and optimized approaches to reduce loneliness among older adults more effectively.

Applying artificial intelligence (AI) technologies in behavioral interventions offers a promising avenue to overcome these challenges by enabling personalized and scalable solutions [11]. AI, first conceived in 1956, initially relied on symbolic AI, which used logical rules and representations to model human intelligence and problem-solving [12]. Over the decades, AI has evolved significantly, advancing into modern approaches such as machine learning (ML), which develops rules from training data [13], and deep learning (DL), a subset of ML that leverages artificial neural networks to model complex patterns in large-scale data [14,15]. Reinforcement learning (RL), another ML subdomain, integrates concepts from psychology and engineering to enable autonomous learning in dynamic environments [16,17]. Robotics, including human–computer interaction (HCI), further extends AI’s applications by addressing physical and emotional needs through interactive technologies [18]. Robotics focuses on designing and building machines that can perform tasks traditionally requiring human intervention [19], and HCI explores the interfaces and interactions between humans and computers to improve usability and accessibility [20]. These capabilities provide scalable solutions to support independent living and enhance the quality of life and mental health of individuals with disabilities and chronic illnesses, including older adults [21]. For this population, AI has enabled improved monitoring of social isolation, the development of companion robots tailored to emotional and social needs, and personalized interventions that foster engagement and psychological well-being [22]. Increasingly, these applications are being leveraged to address loneliness among older adults.

AI-based behavioral interventions take many forms, including social technologies, intergenerational programs, support groups, pet companions, recreational activities, psychological therapies, physical exercises, and assistive technologies [23,24,25,26]. For example, Badal et al. employed natural language processing (NLP) and machine learning models to predict loneliness in older adults using transcribed speech data with high sensitivity and accuracy [3]. Beyond prediction, AI can actively reduce loneliness through innovative tools. Examples include social robots that provide companionship, virtual assistants that simulate conversations and facilitate connections with loved ones, wearable devices that deliver timely interventions, and algorithms that enhance the personalization of interventions [27]. For instance, early social robots like Paro utilized symbolic AI to simulate human-like interactions. Over time, robots have evolved to incorporate more advanced AI capabilities, enabling a broader range of functions and enhanced adaptability to individual needs. These approaches effectively address common access barriers by eliminating the need for travel, offering 24/7 availability, and supporting multiple languages through NLP capabilities. They are generally more cost-effective and may reduce the stigma associated with seeking help from human providers.

Emerging innovations reported in the media further illustrate AI’s potential in reducing loneliness. For instance, an AI-powered board game developed by Johns Hopkins University students combines elements of NLP and interactive gameplay to engage older adults in retirement communities. Modeled after the board game “Guess Who?”, this prototype allows users to converse with an AI opponent while playing, fostering companionship and mental stimulation [28]. Advancements like these highlight the versatility of AI in addressing loneliness through novel, engaging, and scalable approaches.

This study presents a systematic review of AI applications for addressing loneliness among older adults. This review is timely, as AI is rapidly evolving and has the potential to enhance the personalization and scalability of interventions for loneliness among older adults. However, the lack of a comprehensive review leaves a critical gap in understanding how AI has been applied in this space. By mapping the current landscape, this study aims to identify gaps and provide actionable insights to guide future research and development in this promising field.

## 2. Materials and Methods

### 2.1. Review Protocol and Registration

This systematic review was not registered on a public platform. However, the review protocol was developed using the Cochrane Handbook for Systematic Reviews of Interventions as the methodological framework and reported following the Preferred Reporting Items for Systematic Review and Meta-Analysis Protocols (PRISMA-P) guidelines [29].

### 2.2. Study Selection Criteria

Studies meeting all the following criteria were included in the review: (1) Study designs—experimental studies (e.g., randomized controlled trials, pre–post interventions, and cross-over trials) that explicitly tested an AI-enabled intervention aimed at reducing loneliness; (2) Use of AI—The intervention incorporated AI techniques, including symbolic AI, machine learning (ML), deep learning (DL), or reinforcement learning (RL), to intervene in loneliness-related outcomes; (3) Study subjects—Adults aged 55 and older; (4) Outcomes—The primary or secondary goal was to reduce loneliness, including related constructs such as social isolation; (5) Article type—original, empirical, peer-reviewed journal publications; (6) Time window of search—studies published from the inception of electronic bibliographic databases to 31 January 2024; and (7) Language—articles written in English.

Studies were excluded from the review if they met any of the following criteria: (1) studies did not explicitly examine loneliness as an outcome; (2) AI techniques were mentioned but not actively integrated into the intervention (e.g., descriptive discussions of AI without implementation); (3) articles were written in a language other than English; and (4) letters, editorials, study or review protocols, case reports, review articles, or conference abstracts.

### 2.3. Search Strategy

A keyword search was performed in eight electronic bibliographic databases: PubMed/MEDLINE, EBSCO, Academic Search Complete, APA PsycArticles, APA PsycInfo, CINAHL Plus, Web of Science, and Cochrane Library. The search algorithm (Appendix A) includes terms related to AI (e.g., “machine learning”, “neural network”), loneliness (e.g., “social isolation”, “social disconnection”, “emotional isolation”), and older adults (e.g., “aged”, “seniors”, “elderly”) to identify relevant study titles and abstracts in the databases. Two co-authors independently screened titles and abstracts identified from the keyword search, retrieved potentially eligible articles, and evaluated their full texts. Cohen’s kappa (κ = 0.67) was used to assess the interrater agreement between two co-authors. Discrepancies were resolved through discussion. Additionally, relevant articles published after the database search were identified through manual review and included in the study to ensure up-to-date coverage of the topic.

### 2.4. Data Extraction and Synthesis

A standardized data extraction form was used to collect the following methodological and outcome variables from each included study: author(s), year of publication, country/region, study design, overall sample size, arm-specific sample sizes, participants’ age range, participants’ mean/median age, participants’ sex distribution, chronic conditions, AI methodology category, AI implementation fields, intervention type, intervention setting, intervention frequency, intervention duration, nature of intervention, outcome measures, other outcome measures, and intervention effectiveness (i.e., outcome-specific treatment effect estimates).

### 2.5. Study Quality Assessment

The Grading of Recommendations, Assessment, Development, and Evaluations (GRADE) framework was used to assess the quality of each study. This framework evaluates evidence based on factors such as risk of bias, imprecision, inconsistency, indirectness, and publication bias, assigning studies to one of four levels: very low, low, moderate, or high quality. Randomized controlled trials typically begin at high quality, while observational studies start at low quality. The evidence level is adjusted during evaluation based on the presence or absence of these factors, ensuring a transparent and systematic approach to assessing study quality.

## 3. Results

### 3.1. Identification of Studies

Figure 1 shows the PRISMA flow diagram. A total of 223 articles were identified through keyword and reference searches, including one article published after the database search. The articles were retrieved from the following databases: PubMed (102), Web of Science (43), Cochrane Library (16), and EBSCO (including Academic Search Complete, APA PsycArticles, APA PsycInfo, CINAHL Plus, and MEDLINE) (61). After removing duplicates, 148 unique articles underwent title and abstract screening, from which 108 were excluded. The full texts of the remaining 40 articles were reviewed against the study selection criteria. Of these, 31 articles were excluded due to not meeting the inclusion criteria for study design, outcomes, interventions, or article type. No articles were excluded due to the inability to access full text. Therefore, nine studies were included in the systematic review.

### 3.2. Characteristics of Study Participants

Table 1 summarizes the participants’ characteristics in the nine included studies. Studies were published between 2008 and 2024, with seven published after 2020. Participants were recruited from six countries/regions, including the US (*n* = 4), New Zealand (*n* = 2), Taiwan (*n* = 1), Korea (*n* = 1), England (*n* = 1), and Japan (*n* = 1). The mean and median sample sizes are 32 and 33 participants ranging from 15 to 64. Participants’ age ranged from 50 to 100 years. Males constituted 24.96% of participants on average. Chronic conditions among participants varied across studies, including mild depression, cognitive impairment or dementia, and physical disabilities. Some studies excluded participants with severe cognitive impairment or Alzheimer’s disease, severe loneliness, psychiatric conditions, or recent use of psychiatric drugs.

### 3.3. Characteristics of Interventions

Table 2 summarizes the intervention characteristics of the included studies. Two study designs were adopted: RCTs (*n* = 6) and single-arm pre–post studies (*n* = 3). Among the RCTs, four studies randomized participants to two arms (a treatment arm and a control arm), and two studies randomized participants to three arms (one treatment arm and two control arms). The AI methodologies used in these studies included Symbolic AI (*n* = 4), ML (*n* = 5), and DL (*n* = 5). AI implementation fields covered a wide range of technologies, such as speech recognition (*n* = 6), emotion recognition and simulation (*n* = 5), learning and adaptation (*n* = 4), text-to-speech (TTS) (*n* = 4), NLP (*n* = 4), computer vision (CV) (*n* = 3), vision recognition (*n* = 2), autonomous decision-making (*n* = 2), sound localization (*n* = 2), and sensory processing and response (*n* = 2). Intervention types varied, including social robots (*n* = 6), personal voice assistants (*n* = 2), and digital human facilitators (*n* = 1), which are virtual agents designed to simulate human-like appearance and interaction using advanced technologies such as speech synthesis, emotion simulation, and human-like gestures [35].

Most interventions were conducted exclusively in long-term care facilities (LTCFs) (*n* = 6). The remaining studies were implemented in other settings, including senior centers, independent living facilities, and mixed settings such as the community, retirement communities, and nursing homes. The intensity of interventions ranged from 15 min per day to up to 18 h over two weeks, with overall durations spanning from one week to three months. The average duration of the interventions was 1.78 months. The arm-specific sample sizes varied, with the smallest group having 10 participants and the largest group having 33 participants.

### 3.4. Outcome Measures

Table 3 reports the nature of interventions, primary outcome measures, other outcome measures, and intervention effectiveness from the nine included studies. Most studies measured loneliness by the UCLA loneliness scale or its variants [39]. Other outcome measures include scales for attachment to pets, depression (e.g., GDS-SF), quality of life (e.g., WHO-QOL-OLD), cognitive function, perceived stress, psychological well-being, anthropomorphic interactions, and multidimensional scale of perceived social support.

Intervention effectiveness varied among the studies. Six studies [30,31,33,37,38] reported significant reductions in loneliness using various forms of the UCLA Loneliness Scale, particularly those utilizing social robots. Social robots, such as Paro [40] and PIO [34], stood out due to their ability to foster emotional engagement, provide companionship, and adapt to participants’ needs. For example, Lim observed significant improvements in loneliness and cognitive function with robot-led storytelling and gymnastics sessions [34]. Similarly, Robinson et al. highlighted the emotional benefits of interaction with Paro, noting marked loneliness reductions compared to the control group [37].

Three studies [32,35,36] reported non-significant results, possibly attributed to shorter intervention durations, smaller sample sizes, or limited interaction frequencies. Fields et al. used a participatory arts approach combined with social robots, which yielded only minor changes in loneliness, possibly due to the single-session intervention design [32]. Loveys et al. employed a digital human facilitator but observed no significant impact on loneliness, likely due to the brief one-week intervention duration [35]. Papadopoulos et al. investigated culturally competent AI, which integrates cultural knowledge bases to enable socially assistive robots to adapt their interactions to the cultural backgrounds, values, and preferences of users [36]. Despite the advanced tailoring of interactions, the study found no significant reductions in loneliness among older adults, underscoring the need for further refinement and personalization in AI-based interventions to address diverse cultural and individual needs effectively.

### 3.5. Study Quality Evaluation Results

We assessed the evidence/quality of the studies included in the review using the GRADE framework. As shown in Table 1, four studies were rated as “high” quality and the other five “moderate”. Two primary reasons for a “moderate” rating concern a non-randomized study design (pre–post study) and incomplete reporting of participant characteristics and outcomes. Specifically, the studies rated as “moderate” often lacked detailed reporting on mean age, standard deviation, and percentages of males. These gaps in reporting reduce confidence in the results due to potential biases and limited clarity in the study populations.

## 4. Discussion

This review evaluated the research landscape concerning the application of AI in interventions aimed at reducing loneliness in older adults. Among the nine studies included, six reported statistically significant reductions in loneliness, particularly those using social robots. These robots were noted for their ability to simulate human-like interactions, engage users emotionally, and provide consistent companionship. Some studies also suggested that personalized activities, such as storytelling or memory exercises, may have contributed to improved emotional well-being and reduced loneliness. In contrast, three studies reported non-significant effects, which may have been influenced by factors such as shorter intervention durations or less frequent interactions. For example, interventions lasting only a single session or a brief one-week period may not provide sufficient time for AI-assisted companionship to establish meaningful connections. While the review offers insights into potential patterns, it does not allow for definitive conclusions about why some interventions were more effective than others. Nonetheless, the findings suggest that interventions combining AI-assisted companionship with physical or cognitive activities and incorporating longer durations and frequent, meaningful interactions may hold promise for addressing loneliness in older adults.

Consistent with the literature, our findings support the benefits of social robots. Park et al. highlighted the potential of socially assistive robots such as Pepper, ElliQ, and Hyodol in enhancing emotional well-being and reducing loneliness among older adults [41]. These robots not only provided companionship but also facilitated recreational activities like games and simple conversations, which were particularly beneficial for older adults living alone [41]. Similarly, Shah et al. reviewed digital technology interventions (DTIs) such as videoconferencing tools, sensor-based systems, and social apps, finding mixed evidence regarding their long-term effectiveness [42]. While DTIs showed promise in addressing loneliness, the evidence for sustained effects was limited, particularly in studies with shorter durations and smaller sample sizes [42]. Hoang et al. also emphasized that interventions implemented in long-term care settings demonstrated substantial potential in reducing loneliness, though heterogeneity in intervention types and methodologies posed challenges to generalizability [23]. In addition, previous research on animatronic pets has shown their potential in reducing loneliness among older adults. Tkatch et al. found that animatronic pets not only decreased loneliness but also improved psychological outcomes such as resilience and optimism, particularly for individuals with limited ability to care for live pets [43]. These findings highlight the potential for AI-driven technologies to meet diverse needs while minimizing the challenges of traditional pet ownership.

Building upon the success of animatronic pets, humanoid robots represent a more advanced evolution in AI-driven companionship. These robots are designed to mimic human appearance and behavior, enabling more intuitive and engaging interactions [44]. Equipped with speech recognition, natural language processing, and emotion recognition capabilities, humanoid robots can engage in meaningful dialogues, perform tasks such as medication reminders or item retrieval, and provide emotional support by recognizing and responding to human emotions [45]. Their ability to offer personalized companionship makes them especially promising in addressing loneliness. Moreover, their decreasing costs—such as Tesla’s projected $20,000–$30,000 for Optimus or Unitree Robotics’ G1 at $16,000—are making them increasingly accessible [46]. As these technologies continue to advance, humanoid robots are expected to integrate seamlessly into older adults’ lives, further enhancing their potential as tools for fostering companionship and mitigating loneliness.

A critical aspect to consider is the underlying AI technologies employed in these interventions. Speech recognition was the most frequently used technology, enabling conversational engagement and emotional support. For example, Amazon Echo devices utilized speech recognition to facilitate daily interactions, addressing loneliness by providing companionship [33]. Text-to-speech was also employed in Amazon Echo devices, complementing NLP capabilities to create more natural conversational exchanges [33]. Computer vision (CV) was another key technology, aiding robots like PIO in recognizing facial expressions and body language to adapt their responses and foster emotional engagement [34]. Additionally, symbolic AI were integral to social robots like Paro, allowing them to simulate human-like interactions and provide therapeutic support [37]. These technologies collectively enhance the ability of AI interventions to address loneliness by tailoring interactions to individual needs and preferences, highlighting the importance of leveraging advanced AI capabilities in future developments. Moreover, Wu et al. explored the use of conversational agents equipped with advanced NLP and contextual understanding, demonstrating that these agents can facilitate meaningful dialogue with older adults and provide emotional support comparable to human interactions [47]. Similarly, a recent exploratory study found that ChatGPT-3.5 provided emotional support and companionship to older adults, with participants describing it as engaging and easy to use, suggesting its potential as a tool for mitigating loneliness [48]. This suggests that even non-robotic AI technologies may have a role in addressing loneliness.

Our findings echo those from qualitative studies not included in this review. For instance, a qualitative analysis of older adults’ experiences with virtual social interactions via robots during the COVID-19 pandemic found that participants valued the companionship and emotional connection provided by AI robots, describing them as “comforting” and “engaging [49]”. Similarly, Wang et al. highlighted the ability of AI robots to foster a sense of routine and purpose through daily interactions, particularly for those living in isolation [50]. These studies emphasize the importance of emotional engagement and context-specific adaptability in AI interventions. While current qualitative findings highlight the potential benefits of AI-driven companionship, they rely on self-reported experiences, which may introduce variability and subjectivity. Integrating more robust quantitative analyses, such as confidence intervals and statistical effect sizes, would help confirm their impact and generalizability. Existing studies have already demonstrated promise in specific contexts. For example, AI-driven interventions have shown particular promise during solitary leisure activities, such as reading or watching TV, where they can provide conversational engagement and emotional support [51,52]. Additionally, AI robots can guide users through relaxation exercises, cognitive games, or physical activities in therapeutic settings, fostering a sense of connection and well-being [53].

This review was conducted following Cochrane guidance to ensure methodological rigor and systematicity. However, one limitation of our review is the small sample sizes in included studies and the heterogeneity in the duration and intensity of the intervention, which may compromise our findings’ external validity. Another limitation is that our review did not include results from qualitative studies, such as thematic analyses of interview transcripts from participants after using AI robots. These insights could provide valuable context and a deeper understanding of participants’ experiences and perceptions. The overall low percentage of male study participants may also introduce a potential gender bias. Male participants may respond differently to AI interventions due to varying social behaviors, preferences for interaction styles, or different acceptance levels toward technology. This could influence the study outcomes and underscores the need for future research to explore gender-specific responses to AI-based loneliness interventions. Selection bias in the included studies may affect our results, as participants who agreed to enroll may be more open to social engagement [54]. Additionally, some included studies provided limited participant information, such as cognitive impairment status, which might affect the study outcomes. Moreover, studies that did not explicitly use the terms “artificial intelligence” or other AI-specific terminology were unlikely to have been picked up by our search algorithm. For instance, programs focused on conversational agents, chatbot-assisted interventions, and virtual reality may exist but may not have been captured due to the specificity of the search terms. While this review included two studies involving the Paro robot, other Paro-related studies may not have been captured for similar reasons or because they did not meet our inclusion criteria.

This systematic review underscores the promising role of AI in addressing loneliness among older adults, paving the way for further innovation in AI-driven interventions to improve social and emotional well-being. As AI technologies continue to evolve, it is essential to integrate findings from both quantitative and qualitative studies to develop interventions that are not only effective but also deeply personalized and contextually relevant. Future research should focus on scalable models, such as cloud-based platforms that can deliver AI-driven companionship to many users simultaneously, and culturally competent approaches, such as designing AI robots with region-specific languages and customs to resonate with diverse populations. Long-term solutions, including year-round interactive programs that combine AI interventions with human support, should also be explored to ensure that AI interventions are both equitable and impactful in reducing loneliness among older adults.

## 5. Conclusions

AI-driven interventions show promise in mitigating loneliness among older adults by enabling personalized, convenient, and cost-effective support. Social robots in particular—enhanced by emotional simulation, speech recognition, and adaptive capabilities—demonstrate the greatest potential for fostering meaningful engagement. However, further exploration using larger samples, longer intervention periods, and more frequent interactions is needed to refine efficacy and address the complex emotional needs of diverse older populations. Culturally tailored solutions and comprehensive evaluation methods that integrate qualitative and quantitative data may optimize engagement and outcomes. Ultimately, sustainable and scalable AI-based strategies, aligned with ethical and human-centered design principles, are critical for the long-term reduction of loneliness in older adults.

## Figures and Tables

**Figure 1 healthcare-13-00446-f001:**
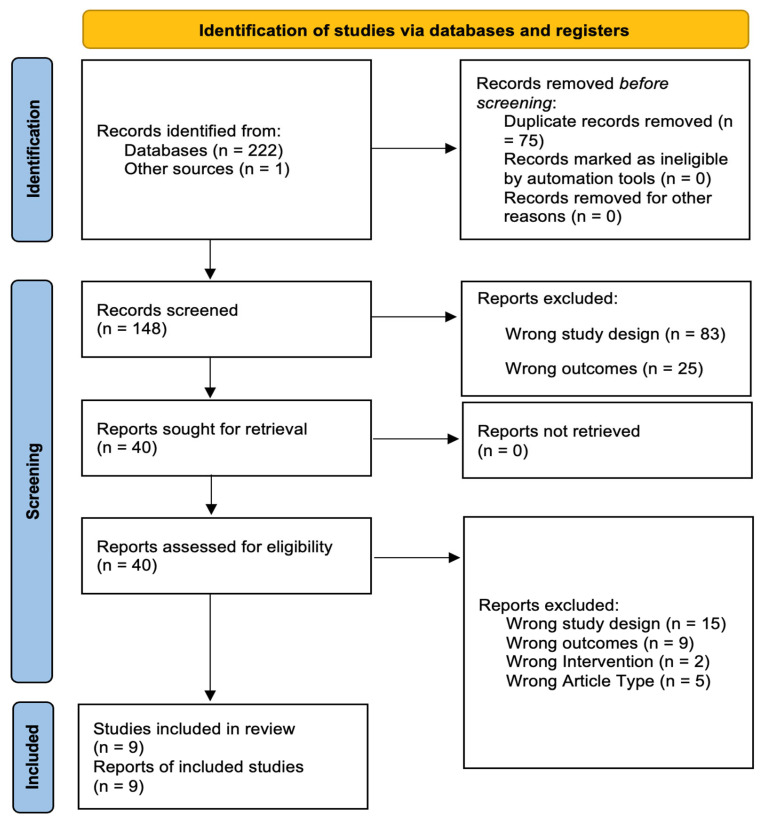
PRISMA flow diagram.

**Table 1 healthcare-13-00446-t001:** Characteristics of the studies included in the review.

Study ID	Author, Year	Country or Region	Study Design	Sample Size	Age Range (Years)	Mean Age ± SD * (Years)	% Males	Chronic Conditions	Grade Score
1	Banks, 2008 [30]	USA	RCT *	38	N/A *	N/A	N/A	Exclude severe cognitive impairment, severe loneliness, psychiatric diseases, or Alzheimer’s disease	Moderate
2	Chen, 2020 [31]	Taiwan	Pre–post	20	65–93	81.1 ± 8.2	35.0%	75% mild depression	Moderate
3	Fields, 2021 [32]	USA	Pre–post	15	77–92	85.80 ± 4.47	26.7%	46.6% dementia	Moderate
4	Jones, 2021 [33]	USA	Pre–post	16	77–96	85.2 ± 5.02	31.0%	N/A	Moderate
5	Lim, 2023 [34]	Korea	RCT	64	65–90	Treatment: 70–74; Control: 75–79	11.0%	20% Physical disability. Exclude history of psychiatric drug, having difficulty using their hands	High
6	Loveys, 2021 [35]	New Zealand	RCT	30	18–69 (8 people); 70+ (22 people)	N/A	20.0%	18–69: with an underlying medical condition; 70+: Mini-Mental State Examination score of >24	Moderate
7	Papadopoulos, 2022 [36]	England and Japan	RCT	33	65–98	81.9 ± 9.82	33.3%	N/A	Moderate
8	Robinson, 2013 [37]	New Zealand	RCT	40	55–100	N/A	N/A	48% cognitive impairment	Moderate
9	Jones, 2024 [38]	USA	RCT	34	50–98	77 ± 11.52	38%	N/A	High

* RCT: randomized controlled trial; SD: standard deviation; N/A: unavailable.

**Table 2 healthcare-13-00446-t002:** Intervention characteristics of the studies included in the review.

Study ID	Author, Year	AI Methodology Category	AI Implementation Fields	Intervention Type	Intervention Setting	Intervention Frequency (/Week)	Intervention Duration (Months)	Arm-Specific Sample Size
1	Banks, 2008 [30]	Symbolic AI, ML *	Speech recognition, vision recognition, autonomous decision-making, emotion recognition and simulation, learning and adaptation	Treatment arm: social robot Control arm1: living dog Control arm2: did not receive the intervention	LTCF *	30 min	2	Treatment arm: 12 Control arm1: 13 Control arm2: 13
2	Chen, 2020 [31]	Symbolic AI	Sensory processing and response, emotion recognition and simulation, learning and adaptation	Single arm: social robot	LTCF	Have 24-h access	2	Single arm: 20
3	Fields, 2021 [32]	ML, DL *	CV *, NLP *, speech recognition, sound localization	Single arm: social robot	LTCF	30 min	Once	Single arm: 15
4	Jones, 2021 [33]	ML, DL	NLP, speech recognition, TTS *	Single arm: Amazon Echo personal voice assistant	LTCF	Avg 18 times daily (first 4 weeks), 10 times daily (next 4 weeks)	2	Single arm: 16
5	Lim, 2023 [34]	Symbolic AI	Speech recognition, emotion recognition and simulation	Treatment arm: social robot Control arm: no intervention	Senior centers	50 min × 2	1.5	Treatment arm: 31 Control arm: 33
6	Loveys, 2021 [35]	DL	Speech recognition, emotion recognition and simulation, learning and adaptation, TTS	Treatment arm: digital human facilitator Waitlist control arm: received the intervention 1 week later.	Conducted remotely, with participants recruited from mixed settings including community-dwelling, independent living, retirement communities, or nursing homes	15 min/day	1 week	Treatment arm: 15 Waitlist control arm1: 15
7	Papadopoulos, 2022 [36]	ML, DL	Speech recognition, vision recognition, autonomous decision-making, emotion recognition, CV, NLP, sound localization, and TTS	Treatment arm: culturally competent Pepper robot; Control arm1: limited cultural competence Pepper robot; Control arm2: care as usual	LTCF	Up to 18 h across 2 weeks	2 weeks	Treatment arm: 12 Control arm1: 11 Control arm2: 10
8	Robinson, 2013 [37]	Symbolic AI	Sensory processing and response, emotion recognition and simulation, learning and adaptation	Treatment arm: interaction with Paro robot; Control arm: attendance at normal activities	LTCF	2 times	3	Treatment arm: 17 Control arm: 17
9	Jones, 2024 [38]	ML, DL	NLP, CV, speech recognition, TTS	Treatment arm: video-based personal voice assistant; Control arm: audio-based personal voice assistant	Independent living facilities	30 min or more per day	3	Treatment arm: 16 Control arm: 18

* ML: Machine learning; LTCF: Long-term care facilities; DL: Deep Learning; CV: Computer Vision; NLP: Natural Language Processing; TTS: Text-to-Speech.

**Table 3 healthcare-13-00446-t003:** Intervention effectiveness of the studies included in the review.

Study ID	Author, Year	Nature of Intervention	Outcome Measures	Other Outcome Measures	Intervention Effectiveness (Mean ± SD)
1	Banks, 2008 [30]	The intervention consisted of animal-assisted therapy sessions where participants interacted with either a robotic dog or a living dog, aimed at reducing loneliness among elderly residents of LTCF *.	UCLA-3 *	Modified Lexington Attachment to Pets Scale	*Within-arm comparisons (baseline* vs. *2 months):* Robotic dog arm: UCLA-3: Significant reduction (*p* < 0.05). Living dog arm: UCLA-3: Significant reduction (*p* < 0.05). Control arm: UCLA-3: No significant changes. *Between-arm comparisons:* UCLA-3: Both robotic dog and living dog resulted in statistically significant improvements in loneliness scores when compared with the control group (*p* < 0.01); No significant difference between the robotic dog and living dog arms.
2	Chen, 2020 [31]	The intervention consisted of each participant being given a Paro (Personal Assistive Robot), aimed to provide companionship and improve mental well-being among older adults with depression living in LTCF.	UCLA-3	GDS-SF *, WHO-QOL-OLD *	*Within-arm comparisons (a week before the start of the 8-week observation (T1)* vs. *immediately at the end of the 8-week observation (T2)* vs. *at the mid-point of the intervention (T3)* vs. *immediately at the end of the intervention (T4)):* UCLA-3: 36.45 vs. 33.8 vs. 23.75 vs. 20.95. In the observation stage (from T1 to T2), there were no significant changes; in the 8-week 24-h Paro intervention (from T2 to T4), the results revealed significant positive changes.
3	Fields, 2021 [32]	The intervention combined participatory arts and social robots, utilizing the social robot and Shakespearean text. Participants interacted with robot, performing Shakespeare’s Sonnet 18 together.	Short form revised UCLA loneliness scale	The Face Scale, GDS-15 *	*Within-arm comparisons (pre-test* vs. *post-test):* Revised UCLA loneliness scale: 4.12 ± 0.92 vs. 3.44 ± 0.91 (*p* = 0.08). *Subgroup comparisons (without dementia* vs. *with dementia):* Changes of Revised UCLA loneliness scale: −0.62 ± 1.04 vs. −0.60 ± 0.55 (*p* = 0.97).
4	Jones, 2021 [33]	Use of Amazon Echo personal voice assistants for conversational interactions aimed at reducing loneliness among older adults.	Abridged eight-item UCLA loneliness scale	Anthropomorphic interactions with the personal voice assistants	*Within-arm comparisons (pre-test* vs. *4 weeks):* Reductions in abridged eight-item UCLA loneliness scale: 2.22 ± 0.42 vs. 1.99 ± 0.45 (*p* = 0.01).
5	Lim, 2023 [34]	The intervention consisted of 12 sessions engaging with robot PIO through storytelling and gymnastics to foster closeness and emotional engagement, transitioning from video-led to robot-led activities over 30-min sessions, concluding with participant reflections.	Russel Revised UCLA loneliness scale	Cognition function, depression, quality of life, mobility, self-care, usual activities, pain/discomfort, anxiety/depression	*Within-arm comparisons (baseline* vs. *6 weeks):* Treatment arm: Russel Revised UCLA loneliness scale: 42.45 ± 11.69 vs. 31.19 ± 11.05 (Change: −11.26 ± 9.34). Control arm: Russel Revised UCLA loneliness scale: 42.03 ± 10.26 vs. 41.48 ± 10.65 (Change: −0.55 ± 10.64). *Between-arm comparisons:* Russel Revised UCLA loneliness scale: t = −4.27 (*p* < 0.01).
6	Loveys, 2021 [35]	The intervention was conducted with a Digital Human, named “Bella”, developed by Soul Machines Ltd. Bella was designed to engage participants in cognitive behavioral and positive psychology exercises through a website interface. The interaction was aimed to target loneliness, stress, and psychological well-being by encouraging daily engagement for at least 15 min over one week.	UCLA-3	Perceived Stress Scale; Flourishing Scale for psychological well-being; Scale of Positive and Negative Experiences for positive and negative affect	*Within-arm comparisons:* No significant main effect of time (F_2,40_ = 0.87; *p* = 0.43; ηp^2^ = 0.04) on perceived loneliness No significant main effect of chronic health condition (F_1,20_ = 0.87; *p* = 0.36; ηp^2^ = 0.04) on perceived loneliness. No significant interaction effect between time and condition on perceived loneliness (F_2,40_ = 0.01; *p* = 0.99; ηp^2^ = 0.00).
7	Papadopoulos, 2022 [36]	The intervention involved the use of culturally competent AI embedded in Pepper robots to support the well-being of older adults in care homes. There were two versions of the AI: a fully culturally competent system (treatment arm) and a more limited version (control arm1). Control arm2 received care as usual without the robot.	ULS-8 *	SF-36 *, CCATool-Robotics *	*Within-arm comparisons (baseline* vs. *8 weeks):* Treatment arm: ULS-8 Score: 14.90 ± 4.98 vs. 14.30 ± 3.53 (*p* > 0.05). Control arm 1: ULS-8 Score: 18.80 ± 4.73 vs. 17.20 ± 6.46 (*p* > 0.05). Control arm 2: ULS-8 Score: 15.70 ± 4.73 vs. 16.50 ± 5.40 (*p* > 0.05). *Between-arm comparisons:* Treatment Arm vs. Control arm 2: ULS-8 Score: No significant improvement (*p* = 0.280). (Treatment & control arm1) vs. Control 2: No significant improvement. Slight improvements in ULS-8 scores, not statistically significant (*p* = 0.290).
8	Robinson, 2013 [37]	The intervention consisted of interaction sessions with Paro aimed at reducing loneliness among elderly residents in LTCF.	UCLA-3	GDS, QoL-AD *	*Within-arm comparisons (baseline* vs. *8 weeks):* Treatment Arm: UCLA-3: 36.44 ± 9.76 vs. 32.23 ± 9.92 (Change: −5.38 ± 7.58) Control Arm: UCLA-3: 31.71 ± 9.50 vs. 33.93 ± 8.52 (Change: +2.29 ± 6.19) *Between-arm comparisons:* There was a significant difference between groups in loneliness change over time (*p* = 0.033).
9	Jones, 2024 [38]	AI-powered personal voice assistants (audio and video) were used for daily engagement to reduce loneliness and improve social support.	UCLA-3	Multidimensional Scale of Perceived Social Support (MSPSS)	*Within-arm comparisons:* Treatment Arm (video-based personal voice assistant): The median change score of perceived loneliness: −0.45 Control Arm (audio-based personal voice assistant): The median change score of perceived loneliness: −0.15 *Between-arm comparisons:* There was a significant difference between groups in loneliness change over time (*p* < 0.05). *Overall (mean ± SD):* Baseline score of perceived loneliness: 1.85 ± 0.61 Post-study score of perceived loneliness: 1.65 ± 0.57

* LTCF: Long-term care facilities; UCLA-3: UCLA Loneliness Scale (Version 3); GDS-SF: the Geriatric Depression Scale-Short Form; WHO-QOL-OLD: the World Health Organization Quality of Life Questionnaire; GDS-15: the 15-item Geriatric Depression Scale; ULS-8: Short form UCLA Loneliness Scale; SF-36: health-related quality of life; CCATool-Robotics: perceptions of robotic cultural competence; QoL-AD: Quality of life in Alzheimer’s disease.

## Data Availability

No new data were created.

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
