# Peer review of "AI Applications to Reduce Loneliness Among Older Adults: A Systematic Review of Effectiveness and Technologies"

_healthcare, 2025, doi:10.3390/healthcare13050446_

Round 1

Reviewer 1 Report

Comments and Suggestions for Authors

The title of the paper clearly expresses the focus of the study. However, the findings section of the abstract should be more detailed and supported by numerical results.

The literature review is comprehensive enough, but some key studies may not have been referenced. More recent literature could be integrated into the study.

The PRISMA flowchart is clear and comprehensive. However, the fact that the protocol was not previously registered on a platform somewhat reduces methodological reliability.

The nine included studies limit the scope of the paper. A larger dataset could improve the generalizability of the results.

The findings highlighted the effect of social robots on loneliness, but the magnitude of these effects (e.g., effect size values) was not explicitly stated.

Three studies reported no significant effect on loneliness. However, the reasons for this situation (short intervention period, small sample size, etc.) can be discussed in more detail.

Especially in cases where qualitative and quantitative results are evaluated together, margins of error and reliability of results should be explained in more detail.

It is stated that the effect of culturally compatible artificial intelligence applications on loneliness is not significant. However, the reasons behind this finding and the need for further personalization are not sufficiently addressed.

Additional graphs (e.g., effect size comparisons or changes over time) can be used to better understand the findings.

The study is theoretically strong but lacks discussion of the findings in a broader context.

The study is successful in proposing AI solutions for loneliness, but it could benefit from more concrete suggestions for real-world applications of these solutions.

The limitations of the study are clearly stated, but some key issues, such as gender balance, could be emphasized more.

Academic writing conventions are followed, but some sentences are too long and complex. These sentences could be reorganized to improve fluency.

Reviewer 2 Report

Comments and Suggestions for Authors

Well organized framework

Also give the experimental setups and arrangements of the devices.

Polish the algorithm / pseudo code with reflect the proposed model.

Ensure all equations are cited problems in the appropriate locations.

Check the journal template correctly formatted and designed in the paper.

All figures and Tables are cited and check figure quality as well.

Share the Dataset access details.

Include the aim, Objectives, Motivation and scope of the proposed work in the introduction section.

Related Works need to be extend with recent published papers. Some Sample related works suggest to refer.

Computational linguistics based text emotion analysis using enhanced beetle antenna search with deep learning during COVID-19 pandemic

- Y Alotaibi, AMS Sundarapandi, P Subhashini, S Rajendran PeerJ Computer Science 9, e1714 Prediction of Attention Deficit Hyperactivity Disorder (ADHD) in Adult using Novel Artificial Neural Network Algorithm

- MS NV, R Surendran 2022 International Conference on Augmented Intelligence and Sustainable Utilizing Hybrid-Deep Learning for Autism Spectrum Disorder Detection in Children via Facial Emotion Recognition

- J Balasubramani, R Surendran 2024 2nd International Conference on Self Sustainable Artificial

Comments on the Quality of English Language

nil

Reviewer 3 Report

Comments and Suggestions for Authors

The article demonstrates internal coherence and is well-structured and substantiated. The introduction is clear, concise, and easy to follow; it addresses the main concepts and provides data that help understand the relevance of the study topic. The results are presented in clear tables, and the discussion highlights the limitations and proposes future lines of research.

My observations focus primarily on the method, results, and discussion sections.

Method:

  • PRISMA Protocol Registration: One of the main requirements of the PRISMA protocol is to ensure the replicability and transparency of a systematic review. What is the reason for not registering it on a public platform? If the article is accepted, would you commit to publishing it?
  • Exclusion Criteria: Exclusion criteria are not simply the opposite of inclusion criteria; they are applied after all inclusion criteria are met. Some exclusion criteria seem to be the opposite of inclusion criteria, rather than complementary. Could you review them to ensure they are appropriate?
  • Search Strategy: There is a lack of justification regarding the selection of databases. Is there any reason for choosing these databases and not others?

Results "3.1. Identification of Studies":

Upon consulting the flow diagram, the following questions arise:

  • How many articles were collected from each database?
  • Did you reject any articles due to inability to access the full text? If so, please detail this information in the text.

I recommend reviewing the PRISMA protocol checklist to ensure compliance with all items required for a systematic review.

Discussion:

Although you address a specific topic, I wonder if a positive view of this intervention has been "emphasized." You mention studies without significant results, relating it to short durations. If there are negative results or controversies on this topic, it would be necessary to address them critically to enrich the study's findings.

Round 2

Reviewer 1 Report

Comments and Suggestions for Authors

Dear Authors,

I have reviewed the revised version you have sent and I see that the updates have been largely taken into consideration. Thank you for your efforts to improve your work and I wish you success.

Best regards

Reviewer 2 Report

Comments and Suggestions for Authors

Accepted now.